# Generation Properties of Stochastic Interpolation under Finite Training Set

## Abstract

This paper investigates the theoretical behavior of generative models under finite training populations. Within the stochastic interpolation generative framework, we derive closed-form expressions for the optimal velocity field and score function when only a finite number of training samples are available. We demonstrate that, under some regularity conditions, the deterministic generative process exactly recovers the training samples, while the stochastic generative process manifests as training samples with added Gaussian noise. Beyond the idealized setting, we consider model estimation errors and introduce formal definitions of underfitting and overfitting specific to generative models. Our theoretical analysis reveals that, in the presence of estimation errors, the stochastic generation process effectively produces convex combinations of training samples corrupted by a mixture of uniform and Gaussian noise. Experiments on generation tasks and downstream tasks such as classification support our theory.

## 1 Introduction

In recent years, generative models have demonstrated remarkable capabilities across various domains, including image generation (Rombach et al., 2022; Peebles & Xie, 2023), image restoration (Wu et al., 2024b;a), text generation (Nie et al., 2025; Ye et al., 2023; Gong et al., 2024), and macromolecular prediction in biomedicine (Trippe et al., 2022). These models aim to learn the target distribution from observed samples and generate new data accordingly. Despite their impressive performance, several studies (Bhattacharjee et al., 2023; Meehan et al., 2020) have reported instances of data-copying behavior in generative models, raising serious concerns about data security, privacy leakage, and copyright infringement. More recent works (Zhang et al., 2023; Yoon et al., 2023; Kadkhodaie et al., 2023; Gu et al., 2023; Somepalli et al., 2023) have systematically examined the memorization behavior of generative models through extensive empirical experiments. These studies reveal that generative models may replicate samples from the training set, exhibiting strong memorization effects, which is particularly prominent with small-sample training sets.

A series of outstanding papers (Oko et al., 2023; Huang et al., 2023) have investigated the theoretical properties of generative models in the asymptotic regime when the number of training samples tends to infinity. However, practical applications often involve finite-sample settings, where the model can only approximate the empirical distribution defined by the training data rather than the true underlying distribution. This raises fundamental questions about the nature and behavior of generative models under such conditions. Several recent works (Gao & Li, 2024; Yi et al., 2023; Bertrand et al., 2025) have taken preliminary steps toward addressing these questions by studying the optimal learning objectives for diffusion and flow-based models in finite-sample regimes and analyzing their generative behaviors.

In this paper, we investigate the memorization and generalization behaviors of generative models through the stochastic interpolation framework(Albergo et al., 2023; Albergo & Vanden-Eijnden, 2022). This framework supports both deterministic and stochastic generation processes, and it encompasses a wide range of diffusion-based approaches, including score-based models (Song et al., 2020) and flow-based models (Lipman et al., 2022). It has also demonstrated strong empirical performance in image generation tasks (Ma et al., 2024;

Yu et al., 2024). As such, the findings of this study offer both broad generalizability and practical relevance.

Our paper analyzes the properties of the stochastic interpolation model under the constraint of limited training samples. We theoretically demonstrate that the model memorizes the training data through its velocity field. Under ideal scenario without estimation error, deterministic generation reproduces the training samples exactly, while stochastic generation produces the training samples with additive Gaussian noise. Furthermore, we investigate the generative behavior under estimation errors and introduce formal definitions of underfitting and overfitting in the context of generative models. These findings lay a theoretical foundation for understanding memorization phenomena in generative modeling.

Contributions of our paper are summarized as follows

- We provide a systematic analysis of the optimal estimation for stochastic interpolation models with finite training populations, and we investigate the theoretical properties of both deterministic and stochastic generation.
- We analyze the effect of estimation errors on generation result and provide a theoretical understanding of overfitting and underfitting phenomena in generative models.
- We reveal that samples generated by stochastic interpolation models can be viewed as training samples perturbed by a combination of uniform and Gaussian noise. We validate our theoretical findings through downstream experiments on datasets such as MNIST, CIFAR-10 and Imagenet.

## 1.1 Notion

In this paper, we adopt the following notions. Let $W_t$ be the standard Brownian motion. $I_d$ denotes the $d$-dimensional identity matrix. The Minkowski sum of two sets is defined as $A * B = \{a + b \mid a \in A, b \in B\}$. The asymptotic notations $f(t) \lesssim g(t)$, $f(t) \gtrsim g(t)$, and $f(t) \asymp g(t)$ indicate that the relationship between $f(t)$ and $g(t)$ is understood up to constant multiplicative factors. The symbol $\delta(x)$ denotes the Dirac delta function. $f'(x)$ represents the derivative of the function $f(x)$ and the gradient operator is denoted as $\nabla$. We use $P_X(\cdot)$ to denote the probability density of the random variable $X$. Specifically, we denote $\rho_t$ as the probability density of the random variable $Z_t$ involved in the differential equation.

## 2 Preliminary: Stochastic Interpolation

In this section we introduce the stochastic interpolation model. Given two random variables $Z_0$ and $Z_1$ with probability distribution $\rho_0$ and $\rho_1$, a stochastic interpolation is defined as

$$Z_t = \mathcal{I}(t, Z_0, Z_1) + \gamma(t)\eta, \ t \in [0, 1], \tag{1}$$

where $\eta$ is the standard Gaussian random variable independent of $Z_0$ and $Z_1$, $\gamma$ is a real-valued function defined on $[0, 1]$ and $\mathcal{I}$ is the function that $[0, 1] \times \mathbb{R}^d \times \mathbb{R}^d \to \mathbb{R}^d$. It has the following conditions.

1. The interpolation function $\mathcal{I}$ satisfies $\mathcal{I}(0, Z_0, Z_1) = Z_0$ and $\mathcal{I}(1, Z_0, Z_1) = Z_1$.

2. The function $\zeta$ satisfies one of the following conditions

- For $t \in \{0, 1\}$, $\gamma(t) \equiv 0$.
- For $t \in (0, 1)$, $\gamma(t) > 0$ and for $\gamma(t) = 0$ elsewhere.

A widely used form of stochastic interpolation is $\mathcal{I}(t, Z_0, Z_1) = \alpha(t)Z_0 + \beta(t)Z_1$, which satisfies $\alpha(0) = 1$, $\alpha(1) = 0$, $\beta(0) = 0$, $\beta(1) = 1$. Therefore, $Z_t$ has the following form

$$Z_t = \alpha(t)Z_0 + \beta(t)Z_1 + \gamma(t)\eta.$$

Without loss of generality, we assume $\alpha(t)$ and $\beta(t) > 0$ when $t \in (0, 1)$. Based on the theoretical frameworks provided by the Fokker–Planck equation and the continuity equation,

the velocity field $b(\boldsymbol{z}, t)$ and the score function $s(\boldsymbol{z}, t)$ are defined as follows

$$b(\boldsymbol{z}, t) = \mathbb{E}\Big[\alpha'(t)Z_0 + \beta'(t)Z_1 + \gamma'(t)\eta | Z_t = \boldsymbol{z}\Big], \tag{2}$$

$$s(\boldsymbol{z}, t) = \nabla_{\boldsymbol{z}} \log \rho_t(\boldsymbol{z}). \tag{3}$$

Starting from an initial point $Z_1 \sim \rho_1$, the generated result obtained from the following generative models satisfy $Z_0 \sim \rho_0$,

$$\text{Deterministic generation: } dZ_t = b(Z_t, t)dt, \tag{4}$$

$$\text{Stochastic generation: } dZ_t = \Big(b(Z_t, t) - \zeta(t)s(Z_t, t)\Big)dt + \sqrt{2\zeta(t)}dW_t. \tag{5}$$

In practical applications, the explicit forms of $b(\boldsymbol{z}, t)$ and $s(\boldsymbol{z}, t)$ are typically unknown and should be estimated through the minimization of the following loss function

$$\hat{b}(\boldsymbol{z}, t) = \arg\min_b \mathbb{E}\Big\|b(Z_t, t) - \Big(\alpha'(t)Z_0 + \beta'(t)Z_1 + \gamma'(t)\eta\Big)\Big\|^2,$$

$$\hat{s}(\boldsymbol{z}, t) = \arg\min_s \mathbb{E}\Big\|s(Z_t, t) - \frac{1}{\gamma(t)}\eta\Big\|^2.$$

The stochastic interpolation model constitutes a general class of generative models. When choosing $\alpha(t) = t$, $\beta(t) = 1 - t$ and $\gamma(t) \equiv 0$, the stochastic interpolation model coincides with the flow mathcing framework. When the initial distribution $\rho_0$ is Gaussian distribution and the time-dependent coefficients $\alpha(t)$ and $\beta(t)$ are suitably selected, the stochastic interpolation model degenerates to the score-based generative model. Therefore, the conclusions discussed in this paper under the stochastic interpolation framework can be readily extended to other diffusion generative models.

## 3 Generation Results under Finite Training Sets

### 3.1 Optimal Generation

In this section, we consider the ideal scenario in which the estimators $\hat{b}$ and $\hat{s}$ are assumed to be error-free, i.e., they exactly coincide with the true underlying functions $b$ and $s$. We begin by analyzing the setting where $\rho_1$ is a Gaussian distribution and $\rho_0$ corresponds to the data distribution, reflecting the classical generative modeling framework. In Section A.1, we will extend our analysis to the more practical case where both $\rho_0$ and $\rho_1$ are empirical distributions constructed from finite populations. The expression of the velocity field $b$, as given in Eq. 2, involves expectations with respect to both $\rho_0$ and $\rho_1$. In many real applications, we are limited to a finite training populations, and therefore $\rho_0$ should be characterized via discretized empirical distributions as follows.

$$\rho_0 = \frac{1}{n}\sum_{i=1}^{n}\delta(X_i). \tag{6}$$

Therefore, taking the expectation with respect to the distribution $\rho_1$ reduces to computing the empirical expectation over the training samples. It is worth noting that we are still taking the expectation with respect to the distribution $\rho_1$, as the Gaussian distribution permits infinite sampling during training. This allows us to obtain the optimal velocity field under finite samples, denoted as $b^*(\boldsymbol{z}, t)$.

Proposition 1 When $\rho_1$ is Gaussian distribution and $\rho_0$ has the discretized empricial distribution form defined in Eq. 6, the optimal velocity field has the following expression

$$b^*(\boldsymbol{z}, t) = \sum_{i=1}^{n} \frac{1}{C_3(t)}\Big[C_1(t)\boldsymbol{z} - C_2(t)X_i\Big] \frac{\exp\big(-\frac{\|\boldsymbol{z} - \alpha(t)X_i\|^2}{2C_3(t)}\big)}{\sum_{j=1}^{n}\exp\big(-\frac{\|\boldsymbol{z} - \alpha(t)X_j\|^2}{2C_3(t)}\big)}, \tag{7}$$

where

$$C_1(t) = \gamma(t)\gamma'(t) + \beta'(t)\beta(t),$$

$$C_2(t) = \Big[\gamma(t)\gamma'(t) + \beta'(t)\beta(t)\Big]\alpha(t) - \Big[\gamma^2(t) + \beta^2(t)\Big]\alpha'(t),$$

$$C_3(t) = \gamma^2(t) + \beta^2(t).$$

$s(\boldsymbol{z}, t)$ can be expressed by $b(\boldsymbol{z}, t)$ and $z$, which is also reported in Huang et al. (2023). Thus, we can obtain the optimal score function under finite populations, denoted as $s^*(\boldsymbol{z}, t)$.

**Proposition 2** When $\rho_1$ is Gaussian distribution and $\rho_0$ has the empirical distribution form defined in Eq. 6, the optimal score function has the expression that

$$s^*(\boldsymbol{z}, t) = \frac{\alpha(t)}{B(t)} b^*(\boldsymbol{z}, t) - \frac{\alpha'(t)}{B(t)} \boldsymbol{z}, \tag{8}$$

where

$$B(t) = \beta(t)\Big[\alpha'(t)\beta(t) - \alpha(t)\beta'(t)\Big] + \gamma(t)\Big[\gamma(t)\alpha'(t) - \gamma'(t)\alpha(t)\Big].$$

It can be observed that the optimal velocity $b^*(\boldsymbol{z}, t)$ has a closded form of a weighted sum of $\{\boldsymbol{z} - X_i\}_{i=1}^n$, which is an expression that can only be derived under finite training sets. Due to the softmax form of the weighting terms, one specific $\boldsymbol{z} - X_i$ will dominate. Therefore there exists $i \in \{1 \cdots n\}$ that the generated results will gradually approach $X_i$, which motivates the following theorem about deterministic generation.

**Theorem 1** There exists $i \in \{1, \ldots, n\}$ such that, based on the optimal velocity field $b^*(\boldsymbol{z}, t)$, the deterministic generated result satisfies $Z_0 = X_i$.

Furthermore, we now consider the case of stochastic generation. Since $s^*(\boldsymbol{z}, t)$ is a linear combination of $b^*$ and $\boldsymbol{z}$, it also gradually becomes a weighted sum of $\{\boldsymbol{z} - X_i\}_{i=1}^n$. This observation motivates the following theorem about stochastic generation.

**Theorem 2** Based on the optimal velocity field $b^*(\boldsymbol{z}, t)$ and score $s^*(\boldsymbol{z}, t)$, assume that $\zeta(t) \lesssim B(t)$ when $t \to 0$, the generated result of stochastic generation defined in Eq. 5 will be the Minikowski sum of the training set and the Gaussian distribution. In particular, there exists $i \in \{1, \ldots, n\}$ such that $Z_0 \sim N(X_i, \sigma^2 I_d)$, where $\sigma^2 = 2 \int_0^t \zeta(t) dt$.

**Remark 1** The assumption that $\zeta(t) \lesssim B(t)$ is satisfied in some common cases, such as $\gamma(t) = \sqrt{t(1-t)}$.

We use a two-dimensional toy data to help understanding and verifying Theorems 1 and 2. Specifically, we randomly select 5 points within the range $[0, 1]^2$ as the training set and compute the corresponding $b^*(\boldsymbol{z}, t)$ and $s^*(\boldsymbol{z}, t)$ using Eq. 7 and 8. The deterministic and stochastic generation results are shown in Fig. 1. When using deterministic generation, the model precisely reproduces the training samples, as the red generated points almost coincide with the blue points in Fig. 1a. In the case of stochastic generation, the generated samples form elliptical Gaussian distributions centered around the training samples. Furthermore, as the noise level increases, the spread of the Gaussian distributions also increases accordingly.

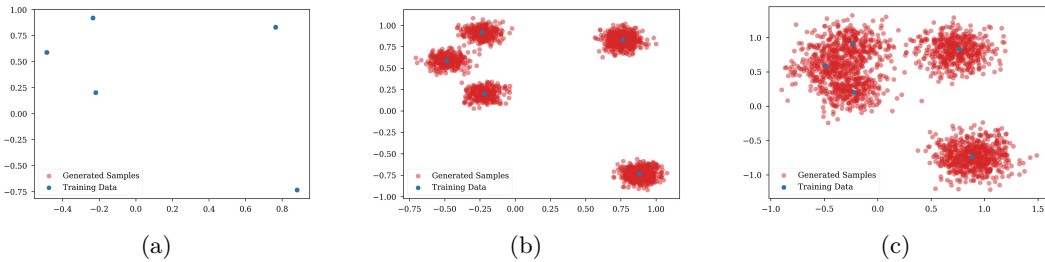

(a)        (b)        (c)

Figure 1: Visualization of oracle generation. Blue points represent the training samples and red points represent the generated samples. (a) Deterministic generation. (b) Stochastic generation with $2 \int_0^1 \zeta(t) dt \approx 0.016$. (c) Stochastic generation with $2 \int_0^1 \zeta(t) dt \approx 0.079$.

## 3.2 Generation with Estimation Error

In practical training, we often adopt various neural networks architectures to estimate the oracle $b^*(\boldsymbol{z}, t)$. Therefore, we further consider the generation process with estimation error. For theoretical simplicity, we focus only on the deterministic generation scenario, since, as analyzed in Section 3.1, stochastic generation can be viewed as deterministic generation with added Gaussian noise.

Let $\hat{b}(\boldsymbol{z}, t)$ denote the estimated velocity field and $\epsilon(\boldsymbol{z}, t)$ denote as the estimation error, i.e.,

$$\epsilon(\boldsymbol{z}, t) := \hat{b}(\boldsymbol{z}, t) - b^*(\boldsymbol{z}, t),$$

which characterizes the discrepancy between the estimated $\hat{b}(\boldsymbol{z}, t)$ and the optimal $b^*(\boldsymbol{z}, t)$. The following theorem reveals the impact of estimation error on the generated results.

**Theorem 3** When using $\hat{b}(\boldsymbol{z}, t)$ as the velocity field and performing deterministic generation defined by Eq. 4, there exists $i \in \{1 \ldots, n\}$ such that the generated result $Z_0$ satisfies

$$Z_0 = X_i - \lim_{t \to 0} \frac{C_3(t)}{C_1(t)} \epsilon(\boldsymbol{z}, t), \tag{9}$$

where $C_1(t)$ and $C_3(t)$ are defined in Proposition 1.

This result indicates that the generated result is a sample from the training set combined with estimation error. In the following, we state formal assumptions on $\epsilon(\boldsymbol{z}, t)$ and investigate its impact on the generation result. The first assumption is the uniform boundness of the error term $\epsilon(\boldsymbol{z}, t)$, which is commonly adopted in convergence analysis (Huang et al., 2023). Under this regularity condition, we establish the following theoretical result.

**Corollary 1** Assume that there exists a constant $\lambda > 0$ such that the approximation error $\epsilon(\boldsymbol{z}, t)$ satisfies

$$\|\epsilon(\boldsymbol{z}, t)\|^2 \leq \lambda, \ \forall \boldsymbol{z}, t \in \mathbb{R} \times [0, 1],$$

Then the generation result of deterministic generation defined by Eq. 4 will lie within the set $\{X_i\}_{i=1}^n$.

Corollary 1 establishes that, provided the error is uniformly bounded, the generated samples remain within the training set, irrespective of the magnitude of the bound. This result follows from the fact that, during the deterministic generation process, the error is explicitly corrected at each iteration. This property highlights a key strength of diffusion-based generative frameworks, namely their robustness to bounded perturbations during the generation trajectory. We visually illustrate the conclusion of Corollary 1 through experiments on two-dimensional data. Setting $\lambda$ to 1, 5, and 25 the generated results with different $\lambda$ are shown in Fig. 2. Even when $\lambda$ is large, the generated results still coincide with the samples in the training set, which supports the conclusion of the Corollary 1.

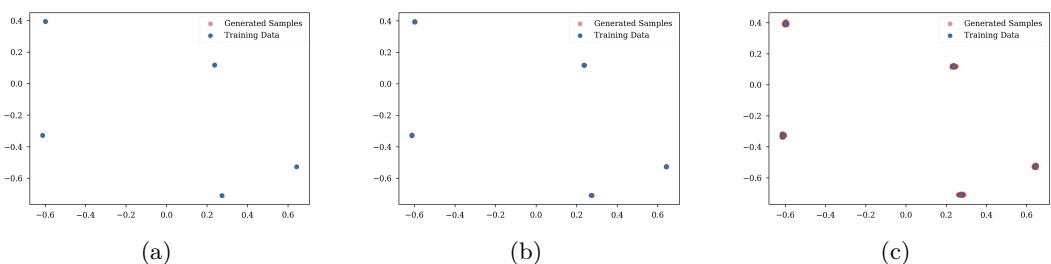

(a)            (b)            (c)

Figure 2: Visualization of the case where the error has a global upper bound. (a), (b), and (c) correspond to $\lambda = 1, 5, 25$ respectively.

The assumption that $\epsilon(\boldsymbol{z}, t)$ is uniformly bounded constitutes a strong condition under which the aforementioned conclusions hold. However, in practice, $\epsilon(\boldsymbol{z}, t)$ typically diverges

due to the divergence of $b^*(\boldsymbol{z}, t)$ as $t \to 0$ or $t \to 1$. To address this, we consider a more general setting in which the divergence of $\epsilon(\boldsymbol{z}, t)$ is regulated by a function $\gamma(t)$. Under this formulation, we demonstrate that the quality and validity of the generation process are governed by the relative decay rates of $\beta(t)$ and $\gamma(t)$ as $t \to 0$.

**Corollary 2** Assume that there exists a constant $\lambda > 0$ such that the estimation error $\epsilon(\boldsymbol{z}, t)$ satisfies

$$\|\epsilon(\boldsymbol{z}, t)\|^2 \le \frac{\lambda}{\gamma(t)}, \ \forall \boldsymbol{z}, t \in \mathbb{R} \times [0, 1].$$

Under the common choice of $\beta(t) = t$, the behavior of the generated samples depends critically on the relative scaling between $\gamma(t)$ and $\beta(t)$ as $t \to 0$:

- If $\gamma(t) \gtrsim \beta(t)$, the generated samples asymptotically converge to the training set.

- If $\gamma(t) \asymp \beta(t)$, the generated samples remain concentrated in the vicinity of the training set.

- If $\gamma(t) \lesssim \beta(t)$, the generated samples diverge, resulting in outputs that deviate substantially from the training set and lack semantic consistency.

In a more general scenario, the generation results are related to the relative scaling between $C_1(t)\gamma(t)$ and $C_3(t)$, as can be seen in Proof. A.3.7.

Corollary 2 offers theoretical guidance for the selection of the functions $\gamma(t)$ and $\beta(t)$ in the deterministic generation process. To develop an intuitive understanding of the implications of Theorem 2, we consider a fixed choice of $\beta(t) = t$ and evaluate three representative forms of $\gamma(t)$: $t(1-t)$, $\sqrt{t(1-t)}$, and $t(1-t)^2$. The corresponding empirical results are presented in Fig. 3.

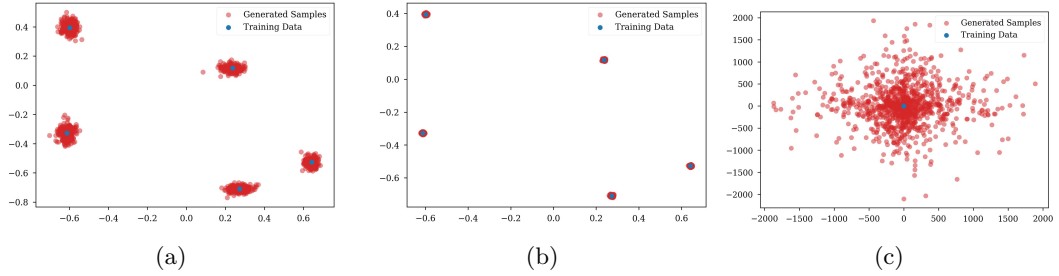

(a)                    (b)                    (c)

Figure 3: Visualization of the case where the error has an upper bound controlled by $\gamma(t)$. (a) $\gamma(t) \asymp \beta(t)$. (b) $\gamma(t) \gtrsim \beta(t)$. (c) $\gamma(t) \lesssim \beta(t)$.

Next, we discuss a more complex but also more realistic error scenario, which is

$$\epsilon(\boldsymbol{z}, t) \propto \frac{1}{\rho_t(\boldsymbol{z})} \propto \sqrt{C_3(t)} \left\{ \sum_{i=1}^n \exp \left( \frac{\|\boldsymbol{z} - \alpha(t) X_i\|^2}{2C_3(t)} \right) \right\}^{-1}.$$

The intuition behind this scenario is that when the marginal density $\rho_t(\boldsymbol{z})$ is relatively high, the likelihood of observing the sample $\boldsymbol{z}$ during training increases, thereby leading to a smaller estimation error at the corresponding pair $(\boldsymbol{z}, t)$. Based on this observation, we assume that the estimation error is inversely proportional to the density $\rho_t(\boldsymbol{z})$. Furthermore, we assume that

$$\epsilon(\boldsymbol{z}, t) = \lambda \left\{ \sum_{i=1}^n \exp \left( -\frac{\|\boldsymbol{z} - \alpha(t) X_i\|^2}{2C_3(t)} \right) \right\}^{-1}, \tag{10}$$

where $\lambda$ is used to control the norm of the estimation error. We neglect the denominator term $\sqrt{C_3(t)\beta(t)}$ in our analysis for the reason that it is a polynomial function of $t$, which is asymptotically dominated by the exponential term in the full expression's denominator.

To streamline the theoretical discussion, we therefore retain only the dominant exponential factor. Moreover, to better reflect practical implementation settings, we adopt the Euler method for discrete-time generation, i.e.

$$Z_{t-1} = Z_t - \hat{b}(\boldsymbol{z}, t)h,$$

where $h$ is the step size. Then, we have the following conclusion.

**Corollary 3** Assume that $\epsilon(\boldsymbol{z}, t) = \lambda \left\{ \sum_{i=1}^{n} \exp\left( -\frac{\|\boldsymbol{z} - \alpha(t)X_i\|^2}{2C_3(t)} \right) \right\}^{-1}$, the generated result of the deterministic method defined in Eq. 4 satisfy

- Underfitting: There exists $\overline{\lambda}$ such that when $\lambda > \overline{\lambda}$, for all $i \in \{1, \ldots, n\}$ and $t$, it holds that

$$\|Z_{t-1} - X_i\|^2 > \|Z_t - X_i\|^2.$$

- Overfitting: Assume the start point in generation is bounded. For any $\tau > 0$, there exists $\underline{\lambda}$ such that when $\lambda < \underline{\lambda}$, there exists $i \in \{1, \ldots, n\}$ satisfying

$$\|Z_0 - X_i\|^2 < \tau.$$

The assumption that the start point is bounded is not particularly restrictive due to the finite numerical precision and constrained sampling ranges in practical implementations. Corollary 3 offers a comprehensive characterization of generative behavior within the stochastic interpolation framework. Specifically, if the estimation error during training is excessively large, the generated samples tend to diverge and fail to capture meaningful structure—this scenario corresponds to underfitting. On the other hand, if the estimation error is extremely small, the generated samples may concentrate too closely around the training data, effectively memorizing them and failing to produce novel outputs, which is referred to as overfitting.

We provide a visual validation of the conclusions drawn in Corollary 3, with the corresponding results presented in Fig. 4. When the error bound $\lambda$ is large, the generated samples exhibit divergent behavior and tend toward infinity, which is the underfitting case. In contrast, when $\lambda$ is sufficiently small, the generated samples remain concentrated around points in the training set, which is the overfitting case. This is consistent with the theoretical predictions.

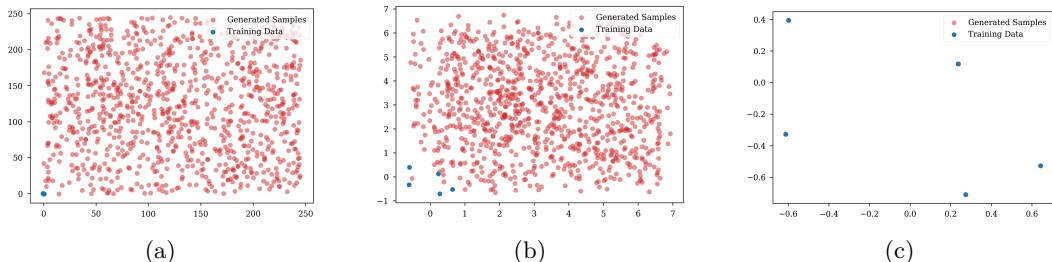

Figure 4: Visualization of the error proportional to the probability: (a), (b), and (c) correspond to $\delta$ values of $1e{-}2$, $1e{-}4$, and $1e{-}10$ respectively.

Section 3.1 reveals that the stochastic generation process inherently injects Gaussian noise into the final outputs. Excluding the underfitting regime, generation with estimation error yields samples that lie within a ball around the training data, which can be effectively interpreted as the addition of uniform noise. Therefore, in practical scenarios, the essence of the generated samples from the generative model is the training samples perturbed by Gaussian noise and uniform noise, where the intensity of Gaussian noise is related to the stochasticity during generation, and the intensity of uniform noise is related to the fitting error.

The conclusions presented in this work do not challenge the fundamental validity of generative models. When the primary objective is sample generation, and the underlying data

space exhibits sufficient continuity, producing samples in the vicinity of the training set can still be considered consistent with the data distribution and may yield high-quality outputs. Furthermore, as the size of the training dataset increases, the generated samples tend to better approximate the true data distribution. Therefore, in classical generative tasks, such as training models on large-scale datasets like ImageNet, it remains feasible to attain both high generation quality and sample diversity.

However, some prior works have utilized samples generated by generative models in downstream tasks, such as training classification models. Our theoretical findings suggest that the use of generated samples in such settings may be effectively equivalent to augmenting the training set with noise-perturbed versions of the original samples. Moreover, when the original training dataset is limited in size, generative models are prone to overfitting, producing samples that closely resemble the training data without introducing substantial novelty. This necessitates caution when applying generative models to downstream tasks.

## 4 Experiments

### 4.1 Performance of Generated Samples in Classification Tasks

In this section, we empirically validate our theoretical findings through experiments on real datasets. Specifically, we assess the effectiveness of generative models in downstream classification tasks using the MNIST and FashionMNIST datasets. To illustrate the sample composition, we consider a total training size of 100 and the training configurations is defined as follow

- Half Samples: A baseline classifier trained on 50 real samples.
- Full Samples: An upper-bound classifier trained on 100 real samples.
- Generated Samples: A classifier trained on 50 real samples and 50 generated samples, where the generative model is trained on the 50 real samples.
- Full Generated Samples: A classifier trained on 50 real samples and 50 generated samples, where the generative model is trained on the full dataset of 60,000 samples.
- Noised Samples: A classifier trained on 50 real samples and 50 noisy versions of them.

The classification accuracies under these settings are reported in Tab. 1 and Tab. 2, where the values in the table represent the classification accuracy (%) on the test set.

Table 1: Classification Accuracy on MNIST

| Sample Size | Half Samples | Noised Samples | Generated Samples | Full Generated Samples | Full Samples |
|---|---|---|---|---|---|
| 100 | 68.84 | 75.3 | 68.61 | 73.74 | 77.57 |
| 200 | 77.57 | 83 | 78.33 | 81.8 | 81.7 |
| 400 | 81.7 | 84.96 | 80.43 | 84.88 | 85.97 |
| 1000 | 87.29 | 88.65 | 86.18 | 88.69 | 89.22 |
| 2000 | 89.22 | 90.44 | 89.78 | 90.48 | 91.3 |
| 4000 | 91.3 | 92.94 | 91.82 | 92.24 | 93.15 |
| 10000 | 93.15 | 94.23 | 93.36 | 93.42 | 94.48 |

Table 2: Classification Accuracy on FashionMNIST

| Sample Size | Half Samples | Noised Samples | Generated Samples | Full Generated Samples | Full Samples |
|---|---|---|---|---|---|
| 100 | 62.26 | 68.69 | 65.89 | 60.8 | 68.32 |
| 200 | 68.32 | 73.78 | 70.78 | 67.81 | 71.04 |
| 400 | 71.04 | 76.18 | 73.07 | 72.5 | 75.77 |
| 1000 | 77.64 | 79.6 | 78.55 | 76.9 | 80.14 |
| 2000 | 80.14 | 82.32 | 81.41 | 80 | 81.83 |
| 4000 | 81.83 | 84.21 | 82.65 | 81.5 | 82.64 |
| 10000 | 84.17 | 84.82 | 83.72 | 83.31 | 84.14 |

The experimental results demonstrate that synthetic samples generated by a generative model trained on the full dataset can enhance the performance of downstream classification tasks, thereby validating the data augmentation potential of high-quality generated samples. However, under small-sample conditions, the limited availability of training data leads the generative model to overfit, resulting in generated samples with low diversity that largely replicate the training distribution. This overfitting behavior is directly manifested in the downstream task performance: specifically, when the training data are scarce, the classification accuracy achieved using generation-augmented data is even lower than that obtained using simple noise-perturbation techniques.

### 4.2 Performance of Generated Samples in Contrastive Learning

We further assess the applicability of our theoretical framework in the context of contrastive learning. Prior work Wang et al. (2024) has shown that incorporating synthetic samples generated by a generative model, together with weaker data augmentation strategies, can enhance contrastive learning performance. Our theory offers a plausible explanation for this observation. Specifically, data augmentation in contrastive learning can be interpreted as generating new samples within a local neighborhood of the original data points. In our theory, the samples generated by the generative model can be regarded as the result of adding Gaussian noise within the unit ball neighborhood of the original training samples. Consequently, combining generated samples with weak augmentation yields a comparable perturbation magnitude to that of conventional strong augmentation strategies (e.g., the $\lambda_{k+1}$ in Wang et al. (2024)), while simultaneously introducing greater sample diversity. This dual effect contributes to improved generalization. We validate this claim through comparions between using generated samples and noised samples under different contrastive learning method on the CIFAR-10 dataset, including SwavCaron et al. (2020), SimsiamChen & He (2021), SimclrChen et al. (2020a) and Mocov2Chen et al. (2020b). Experiments results are reported in Tab. 3, where the values in the table represent the classification accuracy (%) on the test set.

Table 3: Results of Contrastive Learning on CIFAR-10

|  | Swav | Simsiam | Simclr | Mocov2 |
|---|---|---|---|---|
| Baseline | 88.94 | 88.82 | 90.14 | 91.72 |
| Weak Augmentation + Generated samples | 90.11 | 90.80 | 91.71 | 92.95 |
| Weak Augmentation + Noised samples | 90.09 | 90.68 | 91.41 | 92.84 |

The experimental results reveal that incorporating noisy samples in conjunction with weak data augmentation yields modest performance gains over the baseline, although it remains slightly inferior to directly leveraging samples generated by the generative model. This observation is consistent with expectations: the generative model, having been trained on the full dataset of 60,000 samples, possesses strong generative capacity. Nonetheless, the fact that noise based synthetic samples achieve performance comparable to that of the generative model provides indirect empirical support for our theoretical framework.

## 5 Discussion and Future Work

In this paper, we discuss the properties of generative models under limited sample training set based on the stochastic interpolation framework. We also provide a comprehensive understanding for the underfitting and overfitting phenomena in generative models with theoretical support. This conclusion can also be extended to other algorithms in the diffusion model series. However, due to space limitations, there are still some issues that this article does not address, including (1) whether mainstream generative models such as VAE, GAN, and Autoregressive exhibit similar memorization characteristics; (2) whether conditional generative models follow the same patterns under this theoretical framework. These open questions provide valuable directions for future research.

## 6 Reproducibility statement

All experimental results in the paper, including the scatter plots in Section 3 and the tabular data in Section 4, are accompanied by the corresponding code submitted in the supplementary materials. The proofs of all theorems and propositions are provided in Section A.3.

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

## A  Appendix

### A.1  Property of Finite Training Sets on Both End

In this section, we extend the results to the case where both $\rho_0$ and $\rho_1$ are finite sample distributions, which deduces the following proposition.

**Proposition 3** When $\rho_0 = \frac{1}{n}\sum_{i=1}^{n}\delta(X_i)$ and $\rho_1 = \frac{1}{m}\sum_{i=1}^{m}\delta(Y_i)$, the optimal velocity has the following expression

$$b_2^*(\boldsymbol{z}, t) = \frac{\gamma'(t)}{\gamma(t)}\boldsymbol{z} + \sum_{i=1}^{n}\sum_{j=1}^{m}\Big\{\Big[\alpha'(t) - \frac{\gamma'(t)}{\gamma(t)}\alpha(t)\Big]X_i + \Big[\beta'(t) - \frac{\gamma'(t)}{\gamma(t)}\beta(t)\Big]Y_j\Big\}$$

$$\frac{\exp(-\frac{\|\boldsymbol{z}-\alpha(t)X_i-\beta(t)Y_j\|^2}{2\gamma^2(t)})}{\sum_{k=1}^{n}\sum_{l=1}^{m}\exp(-\frac{\|\boldsymbol{z}-\alpha(t)X_k-\beta(t)Y_l\|^2}{2\gamma^2(t)})}. \tag{11}$$

Similar to Theorem 1, we have the following corollary about the generation result.

**Corollary 4** Using $b_2^*(\boldsymbol{z}, t)$ as the velocity field and performing determinstic generation as defined by Eq. 4, starting from a time $t = 1$, the generated result will lie in $\{X_i\}_{i=1}^{n}$. Similarly, starting from time $t = 0$, the generated result will lie in $\{Y_i\}_{i=1}^{m}$.

Corollary 4 provides the result for error-free deterministic generation. Similarly, following the conclusions in Sections 3.1 and 3.2, we can readily obtain the results for stochastic generation and generation with estimation error.

### A.2  The Relation between Generation Quality and Training Set

We further investigate the phenomena of memorization and generalization in generative model training by conducting experiments on the ImageNet dataset. Specifically, we sample 256, 1024, 4096 images randomly from ImageNet to construct the training sets, and adopt the SiT-B/2 Ma et al. (2024) architecture as the backbone of the generative model. To assess the quality of generated images, we employ the CLIPIQA Yang et al. (2022) metric, which provides a reference-free evaluation of perceptual quality, in contrast to metrics such as SSIM and PSNR that require ground-truth comparisons. For quantifying memorization, we follow the evaluation protocol introduced in Yoon et al. (2023). For a generated sample $Z$, find $X_1$ and $X_2$ in the training set with the smallest and second smallest $L_2$ distance. The sample $Z$ is marked as a memorized instance if it satisfies $\frac{\|X_1 - Z\|^2}{\|X_2 - Z\|^2} \leq \frac{1}{3}$ following Yoon et al. (2023). The visual results are presented in Fig. 5.

The results in the figure indicate that if the training set has very few samples, such as 256 samples, the model will quickly memorize the samples in the training set, leading to overfitting. As the number of training samples increases, with 1024 samples, although the quality of the training images improves, the memorization rate also rises. When there are 4096 samples, the model no longer memorizes the samples in the training set and achieves better generation metrics. This memorization phenomenon supports the motivation of this paper.

To further examine this behavior, we display generated samples alongside their closest counterparts in the training set, which is shown in Fig. 5b. The results suggest that, particularly under limited-data settings and prolonged training, the model tends to memorize specific training instances, thereby compromising its generalization capability.

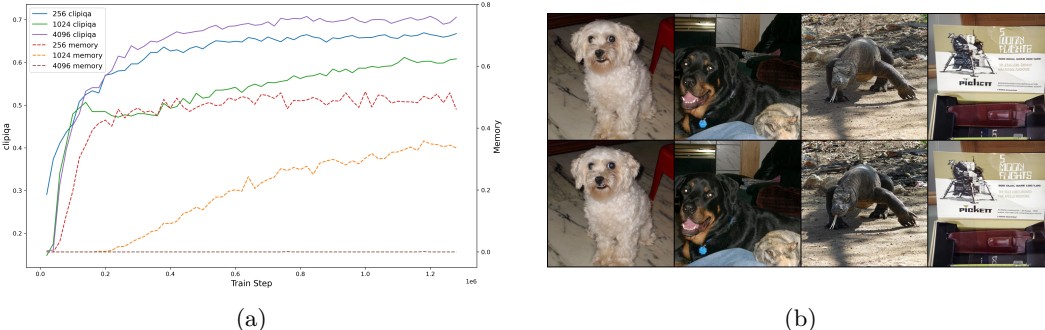

(a)

(b)

Figure 5: Underfitting and overfitting phenomena on ImageNet. (a) Changes in generation quality and memorization as training time increases. (b) Visualization of samples generated by the generative model alongside their closest samples in the training set; the first row shows the generated samples, and the second row shows the closest training samples.

### A.3 Theoretical proofs

In this section, we provide the proofs of the theorems in the main text.

#### A.3.1 Proof of Proposition 1

Proof. According to the definition of $b(\boldsymbol{z}, t)$, when $\rho_1$ is the Gaussian distribution, the optimal velocity field can be derived as

$$
\begin{aligned}
b^*(\boldsymbol{z}, t) &= \mathbb{E}\Big[\alpha'(t)Z_0 + \beta'(t)Z_1 + \gamma'(t)\eta | Z_t = \boldsymbol{z}\Big] \\
&= \frac{1}{p(Z_t = \boldsymbol{z})} \int\int \Big[\alpha'(t)z_0 + \beta'(t)z_1 + \gamma'(t)\eta\Big]\rho_0(z_0)\rho_1(z_1)P_\eta\Big(\frac{\boldsymbol{z} - \alpha(t)z_0 - \beta(t)z_1}{\gamma(t)}\Big)dz_0 dz_1 \\
&= \frac{1}{p(Z_t = \boldsymbol{z})} \int\int \Big\{\big[\alpha'(t) - \frac{\gamma'(t)}{\gamma(t)}\alpha(t)\big]z_0 + \big[\beta'(t) - \frac{\gamma'(t)}{\gamma(t)}\beta(t)\big]z_1 + \frac{\gamma'(t)}{\gamma(t)}\boldsymbol{z}\Big\} \\
&\quad \rho_0(z_0)\rho_1(z_1)P_\eta\Big(\frac{\boldsymbol{z} - \alpha(t)z_0 - \beta(t)z_1}{\gamma(t)}\Big)dz_0 dz_1 \\
&= \Big\{\int\int \rho_0(z_0)\rho_1(z_1)P_\eta\Big(\frac{\boldsymbol{z} - \alpha(t)z_0 - \beta(t)z_1}{\gamma(t)}\Big)dz_0 dz_1\Big\}^{-1} \\
&\quad \int\int \Big\{\big[\alpha'(t) - \frac{\gamma'(t)}{\gamma(t)}\alpha(t)\big]z_0 + \big[\beta'(t) - \frac{\gamma'(t)}{\gamma(t)}\beta(t)\big]z_1 + \frac{\gamma'(t)}{\gamma(t)}\boldsymbol{z}\Big\} \\
&\quad \rho_0(z_0)\rho_1(z_1)P_\eta\Big(\frac{\boldsymbol{z} - \alpha(t)z_0 - \beta(t)z_1}{\gamma(t)}\Big)dz_0 dz_1 \\
&= \Big\{\int\int \rho_0(z_0)\exp\Big(-\frac{\|z_1 - \frac{\beta(t)}{\gamma^2(t)+\beta^2(t)}(\boldsymbol{z} - \alpha(t)z_0)\|^2}{2\frac{\gamma^2(t)}{\gamma^2(t)+\beta^2(t)}}\Big)\exp\Big(-\frac{\|\boldsymbol{z} - \alpha(t)z_0\|^2}{2(\gamma^2(t)+\beta^2(t))}\Big)dz_1 dz_0\Big\}^{-1} \\
&\quad \int\int \Big\{\big[\alpha'(t) - \frac{\gamma'(t)}{\gamma(t)}\alpha(t)\big]z_0 + \big[\beta'(t) - \frac{\gamma'(t)}{\gamma(t)}\beta(t)\big]z_1 + \frac{\gamma'(t)}{\gamma(t)}\boldsymbol{z}\Big\} \\
&\quad \rho_0(z_0)\exp\Big(-\frac{\|z_1 - \frac{\beta(t)}{\gamma^2(t)+\beta^2(t)}(\boldsymbol{z} - \alpha(t)z_0)\|^2}{2\frac{\gamma^2(t)}{\gamma^2(t)+\beta^2(t)}}\Big)\exp\Big(-\frac{\|\boldsymbol{z} - \alpha(t)z_0\|^2}{2(\gamma^2(t)+\beta^2(t))}\Big)dz_1 dz_0
\end{aligned}
$$

Treating $z_1$ as a Gaussian random variable, we obtain

$$\rho_1(z_1)P_\eta\Big(\frac{\boldsymbol{z} - \alpha(t)z_0 - \beta(t)z_1}{\gamma(t)}\Big)$$

$$\propto \exp\Big(-\frac{\|z_1\|^2}{2}\Big)\exp\Big(-\frac{\|\boldsymbol{z} - \alpha(t)z_0 - \beta(t)z_1\|^2}{2\gamma^2(t)}\Big)$$

$$\propto \exp\Big(-\frac{\|z_1\|^2}{2}\Big)\exp\Big(-\frac{\beta^2(t)\|z_1\|^2 - 2\beta(t)z_1^T(\boldsymbol{z} - \alpha(t)z_0) + \|\boldsymbol{z} - \alpha(t)z_0\|^2}{2\gamma^2(t)}\Big)$$

$$\propto \exp\Big(-\frac{(\gamma^2(t) + \beta^2(t))\|z_1\|^2 - 2\beta(t)z_1^T(\boldsymbol{z} - \alpha(t)z_0) + \|\boldsymbol{z} - \alpha(t)z_0\|^2}{2\gamma^2(t)}\Big)$$

$$\propto \exp\Big(-\frac{\|z_1 - \frac{\beta(t)}{\gamma^2(t)+\beta^2(t)}(\boldsymbol{z} - \alpha(t)z_0)\|^2}{2\frac{\gamma^2(t)}{\gamma^2(t)+\beta^2(t)}}\Big)\exp\Big(-\frac{\|\boldsymbol{z} - \alpha(t)z_0\|^2}{2(\gamma^2(t) + \beta^2(t))}\Big).$$

Therefore $b^*(\boldsymbol{z}, t)$ has expression that

$$b^*(\boldsymbol{z},t) = \Big\{\int \rho_0(z_0)\exp\Big(-\frac{\|\boldsymbol{z} - \alpha(t)z_0\|^2}{2(\gamma^2(t) + \beta^2(t))}\Big)dz_0\Big\}^{-1}\int\Big\{\big[\alpha'(t) - \frac{\gamma'(t)}{\gamma(t)}\alpha(t)\big]z_0 + \big[\beta'(t) - \frac{\gamma'(t)}{\gamma(t)}\beta(t)\big]$$

$$\frac{\beta(t)(\boldsymbol{z} - \alpha(t)z_0)}{\gamma^2(t) + \beta^2(t)} + \frac{\gamma'(t)}{\gamma(t)}\boldsymbol{z}\Big\}\rho_0(z_0)\exp\Big(-\frac{\|\boldsymbol{z} - \alpha(t)z_0\|^2}{2(\gamma^2(t) + \beta^2(t))}\Big)dz_0$$

$$= \int\Big[\alpha'(t) - \frac{\gamma'(t)}{\gamma(t)}\alpha(t) - \frac{\beta'(t)\gamma(t) - \gamma'(t)\beta(t)}{\gamma^3(t) + \beta^2(t)\gamma(t)}\alpha(t)\beta(t)\Big]z_0 + \Big[\frac{\gamma'(t)}{\gamma(t)} + \frac{\beta'(t)\gamma(t) - \gamma'(t)\beta(t)}{\gamma^3(t) + \beta^2(t)\gamma(t)}\beta(t)\Big]\boldsymbol{z}$$

$$\frac{\exp\Big(-\frac{\|\boldsymbol{z} - \alpha(t)z_0\|^2}{2(\gamma^2(t)+\beta^2(t))}\Big)}{\int\exp\Big(-\frac{\|\boldsymbol{z} - \alpha(t)z_0\|^2}{2(\gamma^2(t)+\beta^2(t))}\Big)\rho_0(z_0)dz_0}\rho_0(z_0)dz_0.$$

Considering that $\rho_0 = \sum_{i=1}^n \delta(X_i)$, the integral can be written as a summation over the samples $\{X_i\}_{i=1}^n$, thus $b^*(\boldsymbol{z}, t)$ has the following expression

$$b^*(\boldsymbol{z},t) = \sum_{i=1}^n\Big\{\Big[\frac{\gamma(t)\gamma'(t) - \beta'(t)\beta(t)}{\gamma^2(t) + \beta^2(t)}\Big]\boldsymbol{z} - \Big[\frac{\gamma(t)\gamma'(t) - \beta'(t)\beta(t)}{\gamma^2(t) + \beta^2(t)}\alpha(t) - \alpha'(t)\Big]X_i\Big\}\frac{\exp\Big(-\frac{\|\boldsymbol{z}-\alpha(t)X_i\|^2}{2(\gamma^2(t)+\beta^2(t))}\Big)}{\sum_{j=1}^n\exp\Big(-\frac{\|\boldsymbol{z}-\alpha(t)X_j\|^2}{2(\gamma^2(t)+\beta^2(t))}\Big)}$$

$$:= \sum_{i=1}^n\frac{1}{C_3(t)}\Big[C_1(t)\boldsymbol{z} - C_2(t)X_i\Big]\frac{\exp\Big(-\frac{\|\boldsymbol{z}-\alpha(t)X_i\|^2}{2C_3(t)}\Big)}{\sum_{j=1}^n\exp\Big(-\frac{\|\boldsymbol{z}-\alpha(t)X_j\|^2}{2C_3(t)}\Big)},$$

where

$$C_1(t) = \gamma(t)\gamma'(t) + \beta'(t)\beta(t),$$

$$C_2(t) = \Big[\gamma(t)\gamma'(t) + \beta'(t)\beta(t)\Big]\alpha(t) - \Big[\gamma^2(t) + \beta^2(t)\Big]\alpha'(t),$$

$$C_3(t) = \gamma^2(t) + \beta^2(t).$$

$\square$

### A.3.2  Proof of Proposition 2

We begin by presenting two related lemmas that establish some useful properties of $Z_t$ with respect to $Z_0$ and $Z_1$.

Lemma 1 (Tweedie's Formula) Let $Y$ be a random variable following an exponential family distribution, such that $Y = \theta + \varepsilon$, where

- $\theta$ is an unknown parameter follows a prior distribution,

- $\varepsilon$ is independent noise with $\mathbb{E}[\varepsilon] = 0$ and $\text{Var}(\varepsilon) = \sigma^2$.

Then, the posterior expectation of $\theta$ given $Y = \boldsymbol{y}$ is:

$$\mathbb{E}\big[\theta \mid Y = \boldsymbol{y}\big] = \boldsymbol{y} + \sigma^2 \frac{d}{dy} \log P_Y(\boldsymbol{y}).$$

**Lemma 2** The score function $s(\boldsymbol{z}, t)$ can be expressed as

$$s(\boldsymbol{z}, t) = -\frac{1}{\gamma(t)} \mathbb{E}\big[\eta | Z_t = \boldsymbol{z}\big].$$

**Proof.** Recalling that the definition of $Z_t$ is

$$Z_t = \alpha(t) Z_0 + \beta(t) Z_1 + \gamma(t) \eta.$$

Taking conditional expectation with respect to $Z_t$ on both sides, we obtain

$$Z_t = \mathbb{E}\big[\alpha(t) Z_0 + \beta(t) Z_1 | Z_t\big] + \gamma(t) \mathbb{E}\big[\eta | Z_t\big].$$

Applying Lemma 1, it follows that

$$\mathbb{E}\big[\alpha(t) Z_0 + \beta(t) Z_1 | Z_t = \boldsymbol{z}\big] = \boldsymbol{z} + \gamma^2(t) s(\boldsymbol{z}, t).$$

Combining the above equation, we have

$$s(\boldsymbol{z}, t) = -\frac{1}{\gamma(t)} \mathbb{E}\big[\eta | Z_t = \boldsymbol{z}\big]. \tag{12}$$

$\square$

Next we present the proof of Proposition 2.

**Proof.** The velocity field $b(\boldsymbol{z}, t)$ can be further expressed as

$$\begin{aligned} b(\boldsymbol{z}, t) &= \mathbb{E}\big[\alpha'(t) Z_0 + \beta'(t) Z_1 + \gamma'(t) \eta | Z_t = \boldsymbol{z}\big] \\ &= \alpha'(t) \mathbb{E}\big[Z_0 | Z_t = \boldsymbol{z}\big] + \beta'(t) \mathbb{E}\big[Z_1 | Z_t = \boldsymbol{z}\big] + \gamma'(t) \mathbb{E}\big[\eta | Z_t = \boldsymbol{z}\big]. \end{aligned} \tag{13}$$

Applying Lemma 1, it follows that

$$\alpha(t) \mathbb{E}\big[Z_0 | Z_t = \boldsymbol{z}\big] = \boldsymbol{z} + (\beta^2(t) + \gamma^2(t)) s(\boldsymbol{z}, t). \tag{14}$$

In addition, by Lemma 2, we have

$$\begin{aligned} Z_t &= \mathbb{E}\big[\alpha(t) Z_0 + \beta(t) Z_1 | Z_t\big] + \gamma(t) \mathbb{E}\big[\eta | Z_t\big] \\ &= \alpha(t) \mathbb{E}\big[Z_0 | Z_t\big] + \beta(t) \mathbb{E}\big[Z_1 | Z_t\big] + \gamma(t) \mathbb{E}\big[\eta | Z_t\big]. \end{aligned} \tag{15}$$

Combining Eq. 12, Eq. 13, Eq. 14, and Eq. 15, we arrive at the closed-form expression for the score function

$$s(\boldsymbol{z}, t) = \frac{\alpha(t)}{B(t)} b(\boldsymbol{z}, t) - \frac{\alpha'(t)}{B(t)} \boldsymbol{z},$$

where

$$B(t) = \beta(t) \big[\alpha'(t) \beta(t) - \alpha(t) \beta'(t)\big] + \gamma(t) \big[\gamma(t) \alpha'(t) - \gamma'(t) \alpha(t)\big].$$

Thus we have

$$s^*(\boldsymbol{z}, t) = \frac{\alpha(t)}{B(t)} b^*(\boldsymbol{z}, t) - \frac{\alpha'(t)}{B(t)} \boldsymbol{z}.$$

$\square$

### A.3.3 Proof of Theorem 1

Proof. Let us define $A(t) = \exp\left(-\int_t^1 \frac{C_1(u)}{C_3(u)} du\right)$, $X = [X_1, \dots X_n]$ and introduce the weight function

$$\omega(\boldsymbol{z}, t) = \operatorname{softmax}\left[\frac{\|\boldsymbol{z} - \alpha(t)X_1\|^2}{2(\gamma^2(t) + \beta^2(t))}, \cdots, \frac{\|\boldsymbol{z} - \alpha(t)X_n\|^2}{2(\gamma^2(t) + \beta^2(t))}\right]^T.$$

Let $Z_t = A(t)\kappa(t)$, where $\kappa(t)$ is a time-dependent auxiliary function. Then, by differentiating $\kappa(t)$, we have

$$\frac{d\kappa(t)}{dt} = \frac{A(t)\frac{dZ_t(x)}{dt} - Z_t A'(t)}{A^2(t)}$$

$$= \frac{A(t)\frac{C_1(t)}{C_3(t)}Z_t - A(t)\frac{C_2(t)}{C_3(t)} \cdot X \cdot \omega(Z_t, t) - \frac{C_1(t)}{C_3(t)}A_t Z_t}{A^2(t)}$$

$$= -\frac{C_2(t)}{C_3(t)A(t)} \cdot X \cdot \omega(Z_t, t).$$

Let $Z_t(x)$ denote the random variable at time $t$ of the process initialized at point $x$. Then, $Z_t(x)$ can be expressed as follows

$$Z_t(x) = A(t)\kappa(t)$$

$$= \exp\left(-\int_t^1 \frac{C_1(u)}{C_3(u)} du\right)\left(x + \int_t^1 \frac{C_2(u)}{C_3(u)A(u)} \cdot X \cdot \omega(Z_u(x), u) du\right).$$

Therefore, at $t = 0$, there exists $i \in \{1 \cdots n\}$ that the generation result $Z_0(x)$ takes the following value

$$Z_0(x) = \lim_{t \to 0} \exp\left(-\int_t^1 \frac{C_1(u)}{C_3(u)} du\right)\left(x + \int_t^1 \frac{C_2(u)}{C_3(u)A(u)} \cdot X \cdot \omega(Z_u(x), u) du\right)$$

$$= \lim_{t \to 0} \frac{\int_t^1 \frac{C_2(u)}{C_3(u)A(u)} \cdot X \cdot \omega(Z_u(x), u) du}{\frac{1}{A(t)}}$$

$$= \lim_{t \to 0} \frac{\frac{C_2(t)}{C_3(t)A(t)} \cdot X \cdot \omega(Z_t(x), t)}{\frac{A'(t)}{A^2(t)}} \tag{16}$$

$$= \lim_{t \to 0} \frac{C_2(t)}{C_1(t)} \cdot X \cdot \omega(Z_t(x), t)$$

$$= X \cdot \lim_{t \to 0} \omega(Z_t(x), t)$$

$$= X_i.$$

The sixth equation holds because as $t$ approaches 0, due to the nature of the softmax function, one element in $\omega$ tends to 1, while the other elements tend to 0. Therefore, there exists $i \in \{1, \dots, n\}$ such that the final result will be one of the $X_i$.

$\square$

### A.3.4 Proof of Theorem 2

Proof. Recalling that the stochastic generation process has the expression that

$$dZ_t = \left(b^*(Z_t, t) - \zeta(t)s^*(Z_t, t)\right)dt + \sqrt{2\zeta(t)}dW_t,$$

$$\text{where } s(\boldsymbol{z}, t) = \frac{\alpha(t)}{B(t)}b(\boldsymbol{z}, t) - \frac{\alpha'(t)}{B(t)}\boldsymbol{z}.$$

Starting from initial point $s$, $Z_t(x)$ has the following expression

$$Z_t(x) = x - \int_t^1 \left(1 - \frac{\zeta(u)\alpha(t)}{B(u)}\right)b(Z_u(x), u) + \frac{\zeta(u)\alpha'(u)}{B(u)}Z_u(x)du + \int_t^1 \sqrt{2\zeta(t)}dW_t.$$

Let $\tilde{Z}_t$ denote the result generated by the drift component alone (i.e., without the stochastic noise), initialized from $x$

$$\tilde{Z}_t(x) = x - \int_t^1 \left(1 - \frac{\zeta(u)\alpha(u)}{B(u)}\right)b(Z_u(x), u) + \frac{\zeta(u)\alpha'(u)}{B(u)}\tilde{Z}_u(x)du.$$

Similar to the proof of Theorem 1, denoting $A(t) = \exp\left(-\int_t^1 \frac{C_1(u)}{C_3(u)} + \frac{\zeta'(u)\alpha'(u)}{B(u)}du\right)$, we obtain that

$$\tilde{Z}_0(x) = \lim_{t\to 0}\exp\left(-\int_t^1 \frac{C_1(u)}{C_3(u)} + \frac{\zeta'(u)\alpha'(u)}{B(u)}du\right)\left(x - \left(1 - \frac{\zeta(t)\alpha(t)}{B(t)}\right)\int_t^1 \frac{-C_2(u)}{C_3(u)A(u)} \cdot X \cdot \omega(\tilde{Z}_u(x), u)\right)$$

$$= \lim_{t\to 0}\frac{C_2(t)}{C_1(t)}\left(1 - \frac{\alpha'(t)\zeta(t)C_3(t)}{C_1(t)B(t)}\right)^{-1}\left(1 - \frac{\zeta(t)\alpha(t)}{B(t)}\right) \cdot X \cdot \omega(\tilde{Z}_t(x), t).$$

Therefore, when $\zeta(t) \lesssim B(t)$, there exists $i \in \{1, \cdots, n\}$ such that $\tilde{Z}_0(x) = X_i$. Consequently, the stochastic generation process satisfies

$$Z_0(x) = X_i + \int_0^1 \sqrt{2\zeta(t)}dW_t,$$

where $\int_0^1 \sqrt{2\zeta(t)}\,dW_t$ is a Gaussian random variable with mean zero and variance $2\int_0^1 \zeta(t)\,dt$. $\square$

### A.3.5 Proof of Theorem 3

Proof. When an error term $\epsilon(\boldsymbol{z}, t)$ is present, the generation process can be expressed as follows

$$\frac{dZ_t}{dt} = b^*(\boldsymbol{z}, t) + \epsilon(\boldsymbol{z}, t).$$

Analogous to the proof in Theorem 1, define, let $A(t) = \exp\left(-\int_t^1 \frac{C_1(u)}{C_3(u)}du\right)$ and $Z_t = A(t)\kappa(t)$, we have

$$\frac{d\kappa(t)}{dt} = \frac{A(t)\frac{dZ_t}{dt} - Z_t A'(t)}{A^2(t)}$$

$$= \frac{A(t)\frac{C_1(t)}{C_3(t)}Z_t - A(t)\frac{C_2(t)}{C_3(t)} \cdot X \cdot \omega(Z_t, t) + A(t)\epsilon(Z_t, t) - \frac{C_1(t)}{C_3(t)}A_t Z_t}{A^2(t)}$$

$$= -\frac{C_2(t)}{C_3(t)A(t)} \cdot X \cdot \omega(Z_t, t) + \frac{\epsilon(Z_t, t)}{A(t)}.$$

Starting from initial point $x$, $Z_t(x)$ has the following expression

$$Z_t(x) = A(t)\kappa(t)$$

$$= \exp\left(-\int_t^1 \frac{C_1(u)}{C_3(u)}du\right)\left(x - \int_t^1 -\frac{C_2(u)}{C_3(u)A(u)} \cdot X \cdot \omega(Z_u(x), u) + \frac{\epsilon(Z_u, u)}{A(u)}du\right).$$

Therefore, at $t = 0$, there exists $i \in \{1 \cdots n\}$ that the generation result $Z_0(x)$ takes the following value

$$Z_0(x) = \lim_{t \to 0} A(t)\Big(x - \int_t^1 -\frac{C_2(u)}{C_3(u)A(u)} \cdot X \cdot \omega_u(Z_u(x)) + \frac{\epsilon(Z_u, u)}{A(u)} du\Big)$$

$$= \lim_{t \to 0} \frac{\int_t^1 \frac{C_2(u)}{C_3(u)A(u)} \cdot X \cdot \omega(Z_u(x), u) du}{\frac{1}{A(t)}} - \lim_{t \to 0} A(t) \int_t^1 \frac{\epsilon(Z_u, u)}{A(u)} du$$

$$= \lim_{t \to 0} \frac{\frac{C_2(t)}{C_3(t)A(t)} \cdot X \cdot \omega(Z_t(x), t)}{\frac{A'(t)}{A^2(t)}} - \lim_{t \to 0} \frac{\epsilon(Z_t, t)/A(t)}{A'(t)/A^2(t)}$$

$$= \lim_{t \to 0} \frac{C_2(t)}{C_1(t)} \cdot X \cdot \omega(Z_t(x), t) - \lim_{t \to 0} \frac{A(t)\epsilon(Z_t, t)}{A'(t)}$$

$$= X \cdot \lim_{t \to 0} \omega(Z_t(x), t) - \lim_{t \to 0} \frac{C_3(t)}{C_1(t)} \epsilon(Z_t, t)$$

$$= X_i - \lim_{t \to 0} \frac{C_3(t)}{C_1(t)} \epsilon(Z_t, t).$$

The fifth equality holds because $A(t) = \frac{C_1(t)}{C_3(t)} A'(t)$. $\qquad \square$

### A.3.6 Proof of Corollary 1

Proof. Recalling that the expression of $C_1(t)$ and $C_3(t)$ are

$$C_1(t) = \gamma(t)\gamma'(t) + \beta'(t)\beta(t),$$
$$C_3(t) = \gamma^2(t) + \beta^2(t).$$

As $t \to 0$, we observe that $C_3(t)/C_1(t) \to 0$. Therefore, if there exists a constant $\lambda > 0$ such that $\|\epsilon(\boldsymbol{z}, t)\|^2 \leq \lambda$ for all $t$, then

$$\lim_{t \to 0} \left\| \frac{C_3(t)}{C_1(t)} \epsilon(\boldsymbol{z}, t) \right\|^2 = 0.$$

This implies that the error term vanishes in the limit as $t \to 0$, and hence the generated sample at time $t = 0$ concentrates around one of the training samples. Therefore, there exists an $i \in \{1 \cdots n\}$ such that $Z_0 = X_i$. $\qquad \square$

### A.3.7 Proof of Corollary 2

Proof. Since the error term $\epsilon(\boldsymbol{z}, t)$ is bounded by a function related to $\gamma(t)$, if $C_3(t) \lesssim \gamma(t)C_1(t)$, then

$$\lim_{t \to 0} \left\| \frac{C_3(t)}{C_1(t)} \epsilon(\boldsymbol{z}, t) \right\|^2 = 0.$$

If $C_3(t) \asymp \gamma(t)C_1(t)$, there exists a constant $c > 0$ such that

$$\lim_{t \to 0} \left\| \frac{C_3(t)}{C_1(t)} \epsilon(\boldsymbol{z}, t) \right\|^2 = c.$$

If $C_3(t) \gtrsim \gamma(t)C_1(t)$, then

$$\lim_{t \to 0} \left\| \frac{C_3(t)}{C_1(t)} \epsilon(\boldsymbol{z}, t) \right\|^2 = \infty.$$

Next, we demonstrate the case when $\beta(t) = t$, where $\beta'(t) = 1$. we consider the following equation

$$\frac{C_3(t)}{C_1(t)\gamma(t)} = \frac{\gamma^2(t)}{\gamma^2(t)\gamma'(t) + \gamma(t)\beta(t)\beta'(t)} + \frac{\beta^2(t)}{\gamma^2(t)\gamma'(t) + \gamma(t)\beta(t)\beta'(t)}$$

$$= \frac{\gamma^2(t)}{\gamma^2(t)\gamma'(t) + \gamma(t)t} + \frac{t^2}{\gamma^2(t)\gamma'(t) + \gamma(t)t}.$$

Therefore, as $t \to 0$, if $\gamma(t) \lesssim \beta(t)$, then

$$\lim_{t \to 0} \left\| \frac{C_3(t)}{C_1(t)} \epsilon(\boldsymbol{z}, t) \right\|^2 = 0.$$

If $\gamma(t) \asymp \beta(t)$, there exists a constant $c > 0$ such that

$$\lim_{t \to 0} \left\| \frac{C_3(t)}{C_1(t)} \epsilon(\boldsymbol{z}, t) \right\|^2 = c.$$

If $\gamma(t) \gtrsim \beta(t)$, then

$$\lim_{t \to 0} \left\| \frac{C_3(t)}{C_1(t)} \epsilon(\boldsymbol{z}, t) \right\|^2 = \infty.$$

$\square$

### A.3.8 Proof of Corollary 3

Proof. (1) We begin by proving the first case, that is, there exists a constant $\overline{\lambda}$ such that for any $\lambda > \overline{\lambda}$, and for all $i$ and $t$, $\|Z_{t-1} - X_i\|^2 > \|Z_t - X_i\|^2$.

(i) If for all index $j \in \{1 \ldots, n\}$,

$$\|Z_t - \alpha(t)X_j\|^2 > 2 \max_{l,k} \|X_l - X_k\|^2,$$

then, for a fixed index $i$, it follows that for all $j \in \{1 \ldots, n\}$,

$$\begin{aligned}
\left\| Z_t - \alpha(t)X_j \right\|^2 &\leq \left\| Z_t - \alpha(t)X_i \right\|^2 + \alpha(t) \left\| X_i - X_j \right\|^2 \\
&\leq \left\| Z_t - \alpha(t)X_i \right\|^2 + \frac{\alpha(t)}{2} \left\| Z_t - \alpha(t)X_i \right\|^2 \\
&\leq \frac{2 + \alpha(t)}{2} \left\| Z_t - \alpha(t)X_i \right\|^2.
\end{aligned}$$

Consequently, we have

$$\left\{ \sum_{j=1}^n \exp \left( - \frac{\|Z_t - \alpha(t)X_j\|^2}{2C_3(t)} \right) \right\}^{-1} \gtrsim \left\{ n \exp \left( - \frac{\|Z_t - \alpha(t)X_i\|^2}{2C_3(t)} \right) \right\}^{-1} \gtrsim \exp \left( \|Z_t - \alpha(t)X_i\|^2 \right).$$

$$(17)$$

Moreover, there exist universal constants $c_1, c_2 > 0$ such that

$$\begin{aligned}
&\left\| Z_t - \frac{C_1(t)}{C_3(t)} Z_t h + \frac{C_2(t)}{C_3(t)} \cdot X \cdot \omega(Z_t, t)h - X_i \right\|^2 \\
&= \left\| (1 - \frac{C_1(t)}{C_3(t)}h)Z_t - (1 - \frac{C_1(t)}{C_3(t)}h)X_i + (1 - \frac{C_1(t)}{C_3(t)}h)X_i - X_i + \frac{C_2(t)}{C_3(t)} \cdot X \cdot \omega(Z_t, t)h \right\|^2 \\
&\leq (1 - \frac{C_1(t)}{C_3(t)}h) \left\| Z_t - X_i \right\|^2 + \left\| \frac{C_1(t)}{C_3(t)}h X_i \right\|^2 + \left\| \frac{C_2(t)}{C_3(t)} \cdot X \cdot \omega(Z_t, t)h \right\|^2 \\
&\leq c_1 \left\| Z_t - X_i \right\|^2 + c_2.
\end{aligned}$$

$$(18)$$

The constants $c_1$ and $c_2$ are independent of both $t$ and $j$ because the functions $\beta(t)$, $C_1(t)$, $C_3(t)$ and $\|X\|^2$ are all uniformly bounded.

Suppose that the error term $\epsilon(\boldsymbol{z}, t)$ satisfies the following condition

$$\left\| \epsilon(Z_t, t) \right\|^2 h > \left\| Z_t - \frac{C_1(t)}{C_3(t)} Z_t h + \frac{C_2(t)}{C_3(t)} \cdot X \cdot \omega(Z_t, t)h - X_i \right\|^2 + \left\| Z_t - X_i \right\|^2, \qquad (19)$$

then, by virtue of the update equation

$$Z_{t-1} - X_i = Z_t - \frac{C_1(t)}{C_3(t)} Z_t h + \frac{C_2(t)}{C_3(t)} \cdot X \cdot \omega(Z_t, t) h + \epsilon(Z_t, t) h - X_i, \tag{20}$$

and applying the triangle inequality, it follows that

$$\left\| Z_{t-1} - X_i \right\|^2 = \left\| Z_t - \frac{C_1(t)}{C_3(t)} Z_t h + \frac{C_2(t)}{C_3(t)} \cdot X \cdot \omega(Z_t, t) h + \epsilon(Z_t, t) h - X_i \right\|^2$$

$$\geq \left| \| \epsilon(Z_t, t) h \|^2 - \| Z_t - \frac{C_1(t)}{C_3(t)} Z_t h + \frac{C_2(t)}{C_3(t)} \cdot X \cdot \omega(Z_t, t) h - X_i \|^2 \right|$$

$$\geq \| Z_t - X_i \|^2.$$

Combining Eq. 18 and Eq. 19, the condition in Eq. 19 can be rewritten as

$$\| \epsilon(Z_t, t) \|^2 h \geq (1 + c_1) \| Z_t - X_i \|^2 + c_2. \tag{21}$$

According to Eq. 17, for each discrete time $t_i$, there exists a constant $\overline{\lambda}_{t_i}$ such that Eq. 21 holds. We denote $\overline{\lambda}_1 = \max_{t_i} \overline{\lambda}_{t_i}$.

(ii) If there exists an index $j$ such that

$$\| Z_t - \alpha(t) X_j \|^2 \leq 2 \max_{l,k} \| X_l - X_k \|^2,$$

then for any $i \in \{1, \ldots, n\}$, it holds that

$$\| Z_t - \alpha(t) X_i \|^2 \leq \big( 2 + \alpha(t) \big) \max_{l,k} \| X_l - X_k \|^2.$$

In this case, there exist universal constants $c_3 \geq 0$ such that

$$\| Z_t - X_i \|^2 \leq \| Z_t - \alpha(t) X_i \|^2 + \| \alpha(t) X_i - X_i \|^2 \leq c_3,$$

where $c_3$ is independent of both $t$ and $j$. Furthermore, we have

$$\| \epsilon(z, t) \|^2 = \lambda \left\{ \sum_{j=1}^n \exp \left( -\frac{\| Z_t - \alpha(t) X_j \|^2}{2 C_3(t)} \right) \right\}^{-1} \geq \frac{\lambda}{n}.$$

Therefore, there exists a constant $\overline{\lambda}_2$ such that the error term $\epsilon(\boldsymbol{z}, t)$ satisfies Eq. 21. By defining $\overline{\lambda} = \max(\overline{\lambda}_1, \overline{\lambda}_2)$, the proof of the first case is complete.

(2) Next, we prove the second case, that is, for any $\tau > 0$, there exists $\underline{\lambda}$ such that when $\lambda < \underline{\lambda}$, there exists $i \in \{1, \ldots, n\}$ satisfying

$$\| Z_0 - X_i \|^2 < \tau.$$

According to Eq. 20, there exist universal constants $c_1, c_2, c_3 > 0$ such that

$$\left\| Z_{t-1} - X_i \right\|^2 \leq \big( 1 - \frac{C_1(t)}{C_3(t)} h \big) \left\| Z_t - X_i \right\|^2 + \left\| \frac{C_1(t)}{C_3(t)} h X_i + \frac{C_2(t)}{C_3(t)} \cdot X \cdot \omega(Z_t, t)) h \right\|^2$$

$$+ \left\| \epsilon(\boldsymbol{z}, t) \right\|^2 h$$

$$= c_1 \left\| Z_t - X_i \right\|^2 + c_2 + \left\| \epsilon(\boldsymbol{z}, t) \right\|^2 h$$

$$\leq c_3 \lambda \exp(\| Z_t - X_i) \|^2.$$

Since the start point $Z_1$ is bounded and $\{X_i\}_{i=1}^n$ is a finite set, let

$$M = \max_i \| Z_1 - X_i \|^2.$$

Define $C_T := \exp^{(T)}(0)$ as the T-fold exponential, where $T$ is the number of generation steps. Thus there exists $\underline{\lambda}_1$ such that $c_3 \underline{\lambda}_1 M < \ln^{(T)}(C_T)$, which implies $\exp^{(T)}(c_3 \underline{\lambda}_1 M) < C_T$. Simultaneously, there exists $\underline{\lambda}_2$ satisfying $\underline{\lambda}_2 < \frac{\tau}{c_3 C_T}$. By defining $\underline{\lambda} = \min(\underline{\lambda}_1, \underline{\lambda}_2)$, we obtain $c_3 \underline{\lambda} \exp^{(T)}(c_3 \underline{\lambda} M) \leq \tau$, and consequently, $\| Z_0 - X_i \|^2 \leq \tau$, which completes the proof of the second case.

$$\square$$

### A.3.9 Proof of Proposition 3

Proof. Similar to the proof of Proposition 1, when $\rho_0 = \frac{1}{m}\sum_{i=1}^m \delta(X_i)$ and $\rho_1 = \frac{1}{n}\sum_{i=1}^n \delta(Y_i)$, the optimal velocity field has the expression that

$$b_2^*(\boldsymbol{z}, t) = \mathbb{E}\big[\alpha'(t)Z_0 + \beta'(t)Z_1 + \gamma'(t)\eta | Z_t = \boldsymbol{z}\big]$$

$$= \frac{1}{p(Z_t = \boldsymbol{z})} \int\int \big[\alpha'(t)Z_0 + \beta'(t)Z_1 + \gamma'(t)\eta\big]\rho_0(z_0)\rho_1(z_1)P_\eta\Big(\frac{\boldsymbol{z} - \alpha(t)z_0 - \beta(t)z_1}{\gamma(t)}\Big)dz_0 dz_1$$

$$= \frac{1}{p(Z_t = \boldsymbol{z})} \int\int \Big\{\Big[\alpha'(t) - \frac{\gamma'(t)}{\gamma(t)}\alpha(t)\Big]z_0 + \Big[\beta'(t) - \frac{\gamma'(t)}{\gamma(t)}\beta(t)\Big]z_1 + \frac{\gamma'(t)}{\gamma(t)}\boldsymbol{z}\Big\}$$

$$\rho_0(z_0)\rho_1(z_1)P_\eta\Big(\frac{\boldsymbol{z} - \alpha(t)z_0 - \beta(t)z_1}{\gamma(t)}\Big)dz_0 dz_1$$

$$= \frac{\gamma'(t)}{\gamma(t)}\boldsymbol{z} + \sum_{i=1}^n\sum_{j=1}^m \Big\{\Big[\alpha'(t) - \frac{\gamma'(t)}{\gamma(t)}\alpha(t)\Big]X_i + \Big[\beta'(t) - \frac{\gamma'(t)}{\gamma(t)}\beta(t)\Big]Y_j\Big\}$$

$$\frac{\exp\big(-\frac{\|\boldsymbol{z} - \alpha(t)X_i - \beta(t)Y_j\|^2}{2\gamma^2(t)}\big)}{\sum_{k=1}^n\sum_{l=1}^m \exp\big(-\frac{\|\boldsymbol{z} - \alpha(t)X_k - \beta(t)Y_l\|^2}{2\gamma^2(t)}\big)}.$$

$\square$

### A.3.10 Proof of Corollary 4

Proof. According to the definition of $b_2^*(\boldsymbol{z}, t)$ and the proof of Theorem 1, we denote $A(t) = \exp\big(-\int_t^1 \frac{\gamma'(u)}{\gamma(u)}du\big)$. Starting from an initial point $x$ at time $t = 1$, $Z_t(x)$ has expression that

$$Z_t(x) = \exp\Big(-\int_t^1 \frac{\gamma'(u)}{\gamma(u)}du\Big)\Big(x - \int_t^1 \sum_{i=1}^n\sum_{j=1}^m \Big\{\Big[\alpha'(u) - \frac{\gamma'(u)}{\gamma(u)}\alpha(u)\Big]X_i + \Big[\beta'(u) - \frac{\gamma'(u)}{\gamma(u)}\beta(u)\Big]Y_j\Big\}$$

$$\frac{\exp\big(-\frac{\|Z_u(x) - \alpha(u)X_i - \beta(u)Y_j\|^2}{2\gamma^2(u)}\big)}{\sum_{k=1}^n\sum_{l=1}^m \exp\big(-\frac{\|Z_u(x) - \alpha(u)X_k - \beta(u)Y_l\|^2}{2\gamma^2(u)}\big)}\frac{1}{A(u)}du\Big).$$

Therefore, there exists an $i \in \{1, \ldots, n\}$ such that

$$Z_0(x) = \lim_{t\to 0}\exp\Big(-\int_t^1 \frac{\gamma'(u)}{\gamma(u)}du\Big)\Big(\int_t^1 \sum_{i=1}^n\sum_{j=1}^m \Big\{\Big[\alpha'(u) - \frac{\gamma'(u)}{\gamma(u)}\alpha(u)\Big]X_i + \Big[\beta'(u) - \frac{\gamma'(u)}{\gamma(u)}\beta(u)\Big]Y_j\Big\}$$

$$\frac{\exp\big(-\frac{\|Z_u - \alpha(u)X_i - \beta(u)Y_j\|^2}{2\gamma^2(u)}\big)}{\sum_{k=1}^n\sum_{l=1}^m \exp\big(-\frac{\|Z_u - \alpha(u)X_k - \beta(u)Y_l\|^2}{2\gamma^2(u)}\big)}\frac{1}{A(u)}du\Big)$$

$$= \lim_{t\to 0} -\frac{\gamma(t)}{\gamma'(t)}\sum_{i=1}^n\sum_{j=1}^m \Big\{\Big[\alpha'(u) - \frac{\gamma'(u)}{\gamma(u)}\alpha(u)\Big]X_i + \Big[\beta'(u) - \frac{\gamma'(u)}{\gamma(u)}\beta(u)\Big]Y_j\Big\}$$

$$\frac{\exp\big(-\frac{\|z_u - \alpha(u)X_i - \beta(u)YY_j\|^2}{2\gamma^2(u)}\big)}{\sum_{k=1}^n\sum_{l=1}^m \exp\big(-\frac{\|Z_u - \alpha(u)X_k - \beta(u)Y_l\|^2}{2\gamma^2(u)}\big)}$$

$$= \lim_{t\to 0}\frac{-1}{\gamma'(t)}\Big[\sum_{i=1}^n\sum_{j=1}^m \Big(\gamma(t)\alpha'(t) - \gamma'(t)\alpha(t)\Big)X_i + \Big(\gamma(t)\beta'(t) - \gamma'(t)\beta(t)\Big)Y_j\Big]$$

$$\frac{\exp\big(-\frac{\|Z_u - \alpha(u)X_i - \beta(u)Y_j\|^2}{2\gamma^2(u)}\big)}{\sum_{k=1}^n\sum_{l=1}^m \exp\big(-\frac{\|Z_u - \alpha(u)X_k - \beta(u)Y_l\|^2}{2\gamma^2(u)}\big)}$$

$$= X_i.$$

According to the symmetry of $\{X_i\}_{i=1}^n$ and $\{Y_j\}_{j=1}^m$, starting from time $t = 0$, the generated samples belong to $\{Y_j\}_{j=1}^m$.

$\square$

### A.4   LLM Usage Statement

Large Language Models were used solely as an auxiliary tool for translation and language polishing of the manuscript. The research ideas, problem formulation, methodology, analysis, and main writing were entirely conducted by the authors without LLM assistance.

