# OpenReview forum: "Generation Properties of Stochastic Interpolation under Finite Training Set"
_ICLR.cc/2026/Conference — ICLR 2026 Conference Withdrawn Submission_

### Official Review · Reviewer_nhQG · 2025-10-28

**Soundness:** 1
**Presentation:** 2
**Contribution:** 2
**Rating:** 2
**Confidence:** 4

**Summary:**

This paper aims at analyzing the behavior of a class of generative models trained on limited data through the stochastic interpolation framework. It asserts that, under ideal conditions, deterministic generation exactly reproduces the training samples, while stochastic generation corresponds to the same samples perturbed by Gaussian noise. When estimation errors are introduced, the generated samples become convex combinations of training data corrupted by mixed uniform and Gaussian noise. It also proposes definitions of overfitting and underfitting specific to generative models. The theoretical discussion is supported by experiments on MNIST, CIFAR-10, and ImageNet, which are said to confirm that small training sets lead to memorization, while larger datasets yield more diverse generations. Overall, the paper positions itself as providing a theoretical explanation for memorization and generalization phenomena in generative models trained on finite datasets.

**Strengths:**

1. *Link between drift, score, and noise:* The paper explicitly relates the forms of the velocity (drift) and score functions to the type and magnitude of stochastic noise, synthesizing how Gaussian or uniform perturbations emerge in generation.
2. *Characterization of memorization:* It offers a formal description of how deterministic generation reproduces training data and how stochasticity leads to noisy versions, providing a claimed theoretical explanation of memorization phenomena in generative models.
3. *Introduction of overfitting/underfitting notions:* It discusses these regimes specifically for generative models, linking them to the size of estimation errors and to the behavior of the velocity field.
4. *Data augmentation perspective:* The paper includes technical examples and experiments showing how generated samples can act as noise-perturbed or augmented data, connecting generation quality to data availability.
5. *Conceptual contribution:* Overall, it attempts to systematize how finite-sample size, stochasticity, and model estimation errors jointly shape the generative process.

**Weaknesses:**

**Major Scientific Weaknesses**
1. **Poor referencing and limited contextualization:** The paper insufficiently cites the rapidly growing literature on the theoretical convergence and generalization of diffusion-based models. Several key contributions ([1]–[4]) addressing convergence under manifold hypotheses, log-concavity relaxations, and Wasserstein or KL guarantees are omitted. As a result, the theoretical positioning of this work within the broader landscape remains unclear.
2. **Redundant theoretical content:**
The connection between the drift (velocity field) and score function has already been well established in prior studies ([5]–[7], even though not cited), and some of the presented results appear to directly reproduce existing ones. In particular, Proposition 2 is an almost verbatim restatement of Corollary 2.1 in [8], without distinction regarding novelty.
3. **Questionable correctness of results:** Several claimed results seem mathematically imprecise. In particular, the reasoning in Theorem 2 is problematic: it separates the deterministic and stochastic components of an SDE despite the drift being nonlinear, which invalidates the argument under standard stochastic analysis. The claimed derivation of the Gaussian perturbation outcome therefore lacks rigor.
4. **Unrealistic assumptions:** The assumptions on boundedness and smoothness (e.g., uniform or infinite upper bounds on certain norms and distances) are overly restrictive and not justified under realistic neural network approximations. These conditions make the practical relevance of the theoretical claims doubtful.
5. **Vague definition of overfitting and underfitting:** The proposed definitions are not linked to the learning or optimization process—they are defined in terms of generative outcomes rather than training dynamics. Moreover, the chosen error structure lacks justification: it is unclear why such a specific functional form of the estimation error is meaningful or observable in practice.

**Presentation and Structural Problems**
1. **Formatting and compliance:** The manuscript seems not to follow the official ICLR template (e.g., presentation of theorems and propositions), producing an irregular and somewhat confusing layout.
2. **Frequent typographical errors:** Numerous typos and grammatical issues occur throughout (e.g., lines 78, 126, 152, 175, 190, 223, 455, 456–457). The paper clearly requires thorough proofreading and language polishing.
3. **Misleading or unclear figures:**
The toy illustrations do not convincingly relate to actual generative model behavior. It is unclear whether they stem from trained models or analytical constructions. Without methodological details, these visuals risk being pedantic and misleading rather than informative.

While the paper raises a relevant question about the behavior of generative models under finite training data, it lacks precise referencing, relies on overly strong assumptions, and reproduces known results without sufficient methodological rigor or clear novelty. Significant theoretical and presentation revisions are required before it can be considered a reliable contribution.

**Questions:**

See weaknesses

---

### Official Review · Reviewer_atQx · 2025-10-31

**Soundness:** 2
**Presentation:** 2
**Contribution:** 2
**Rating:** 4
**Confidence:** 3

**Summary:**

The paper analyzes memorization/generalization of the Stochastic Interpolation. With one end being empirical measure and the other end being Gaussian, the authors derive closed‑form expressions for the optimal velocity being a softmax‑weighted combination of the training samples, and show the optimal score is an affine transform of
the optimal velocity. Using these forms, they prove that deterministic generation recovers exact training samples, while stochastic generation yields training samples plus Gaussian noise; they then study how estimation error affects outcomes, and present toy examples and tasks (MNIST/FashionMNIST classification; CIFAR‑10 contrastive learning) to support the theory.

**Strengths:**

Overall the paper is clearly written with simple and intuitive main theorems. Thm. 1/2 formalize the ``exact copy vs. noisy copy" intuition, and the visualization in Fig. 1 matches the theory. It is interesting that the author takes into account the estimation error and connect it to the under/overfitting in SI with finite‑data training. The toy studies in Fig. 3–4 qualitatively support the predictions.

**Weaknesses:**

1. The memorization effect and the closed‑form optimal scores in finite‑sample settings are well established in many score‑based diffusion literature. The SI perspective substantially generalize the setting, but the incremental conceptual gain over prior SGM memorization results is limited without new theoretical consequences or scaled experiments on modern, larger models. The paper would benefit from a clearer comparisons of what SI enables beyond standard SGM results.
2. While Section 4 provides two intuitive experiments, the evidence is indirect. Since the paper derives closed‑form optimal estimators for the velocity and score, it would be informative to include generation results obtained with the closed‑form optimal scores as a comparison to the trained model. Additionally, reporting the discrepancy between the learned score function and the theoretical optimal score function would more concretely validate the proposed theory and help isolate the effect of training errors.
3. There are several formatting problems in the manuscript. For example, theorem statements appear inline with the main text without typographic emphasis. Please consider using standard theorem environments. Also there are problem with the citation format in Section 4.2.

**Questions:**

Could the authors clarify what the first column ("Sample Size") in Tables 1 and 2 represents? The paper does not define this term, and it seems inconsistent with the statement in the main text that the training sample size is fixed at 100. If "Sample Size" does not denote the actual number of training examples used, please specify precisely what it measures and how it relates to the experimental setup.

---

### Official Review · Reviewer_DNhD · 2025-10-31

**Soundness:** 1
**Presentation:** 1
**Contribution:** 1
**Rating:** 2
**Confidence:** 5

**Summary:**

This paper addresses the problem of generating samples from a target distribution using the stochastic interpolation method. Unlike previous work, the target distribution here is assumed to be discrete, with its support concentrated on a finite set of points.
The main contributions are theoretical: we demonstrate that both the deterministic and stochastic versions of the generator produce synthetic samples that are either elements of the support (in the deterministic case) or samples formed by adding a Gaussian vector to an element of the support. The case of an estimated score is also examined.

**Strengths:**

Given the current prominence of generative modeling, any effort to deepen our theoretical understanding of existing methods or to design new, theoretically grounded approaches is highly valuable.

**Weaknesses:**

### General Assessment
The paper suffers from significant issues in both language and mathematical rigor, making it difficult to follow. Below is a detailed list of the problems identified:

---

### **1. Language and Mathematical Issues**

- **Lines 76, 78:** The term "Notion" should be replaced with "Notation."
- **Line 83:** The authors state that $P_X$ denotes the probability density function (PDF) of a random variable $X$.
  - First, there is no evidence in the text where this notation is actually used.
  - Second, the reference measure for the density should be explicitly clarified.
  - Instead, the notation $p$ appears on line 681 without any prior definition.
- **Line 97:** The notation $\zeta$ is introduced without explanation. It remains unclear whether $\zeta$ is equivalent to $\gamma$ or refers to a different quantity.
- **Line 107:** The phrase *"Without loss of generality"* lacks justification—it is unclear why the condition does not restrict the generality of the setting.
- **Line 113:** A reference is needed to support the claim that Equations (4) and (5) have solutions and that these solutions possess the desired marginals.
- **Lines 120–123:** While it is acknowledged that $b$ and $s$ are unknown in practice, the text should specify what *is* known to properly define the estimation problem.
  - The formulas in these lines include $Z_t$ on the right-hand side, but it is unclear how $Z_t$ can be accessed when $b$ and $s$ are unknown.
  - The left-hand side depends on $\boldsymbol{z}$, yet $\boldsymbol{z}$ does not appear on the right-hand side.
- **Line 142:** The statement *"In many real applications ..."* is either incorrect or requires reformulation.
- **Line 147:** It is unclear why expectations with respect to $\rho_1$ are treated as empirical means. While this may hold for $\rho_0$ (as defined on line 145), it does not necessarily apply to $\rho_1$.
- **Formatting:** Boldface font should be used consistently for **Theorems**, **Propositions**, etc.
- **Proposition 1:** The meaning of *"optimal"* is ambiguous, rendering the claim unclear.
- **Theorem 3:** In Equation (9), $z$ should be replaced with $Z_0$. Additionally, the existence of the limit should be explicitly stated and proven.

---

### **2. Problem Formulation**
The paper fails to clearly specify the target distribution for sampling. In generative modeling, the typical setup involves:
- An unknown target distribution $P_0$.
- An empirical distribution $\hat{P}_n$ derived from a sample of size $n$ drawn independently from $P_0$.

In this paper, there is confusion between $P_0$ and $\hat{P}_n$. As far as I understand, the authors assume $P_0$ is a uniform distribution over a finite set, with elements denoted by $X_i$. However, the fact that backward ODEs and SDEs lead to elements from the support of $P_0$ was already established in prior work.

---

### **3. Limited Novelty and Impact**
The paper's contributions are modest and of limited practical interest:
- The discretization of deterministic and stochastic differential equations is not addressed, despite its critical impact on the generative model's error.
- The selection of functions $\alpha$ and $\beta$ is not discussed, even though these choices may significantly affect performance.


---

### **4. Experimental Setting**
The experimental section lacks clarity:
- The choice of parameters ($\alpha$, $\beta$, $\gamma$), the discretization scheme, and other experimental details should be explicitly described.
- The theoretical insights that the experiments aim to validate should be clearly stated.

---

### **5. Prior work**
The paper claims to investigate memorization and generalization behavior of genertaive models (line 49). But unfortunatly, it does not contain any discussion of what has been done in the prior work on these aspects. Relevant references include
-  Vaishnavh Nagarajan, Colin Raffel, and Ian J Goodfellow. Theoretical Insights into Memorization in GANs. Neural
Information Processing Systems (NeurIPS) 2017 - Integration of Deep Learning Theories Workshop, 2018.
- Ishaan Gulrajani, Colin Raffel, and Luke Metz. Towards GAN Benchmarks Which require Generalization. In ICLR 2019
- Matthew Jagielski et al.  Measuring Forgetting of Memorized Training Examples. In The Eleventh International Conference on Learning Representations, 2023
- Gowthami Somepalli, Vasu Singla, Micah Goldblum, Jonas Geiping, and Tom Goldstein. Diffusion Art or Digital Forgery? Investigating Data Replication in Diffusion Models. In Proceedings of the IEEE/CVF Conference on
Computer Vision and Pattern Recognition, pages 6048-6058, 2023a.
- Gowthami Somepalli, Vasu Singla, Micah Goldblum, Jonas Geiping, and Tom Goldstein. Understanding and Mitigating Copying in Diffusion Models. In Thirty-seventh Conference on Neural Information Processing Systems, 2023b.
- Elen Vardanyan, Sona Hunanyan, Tigran Galstyan, Arshak Minasyan, Arnak S. Dalalyan:
Statistically Optimal Generative Modeling with Maximum Deviation from the Empirical Distribution. ICML 2024

**Questions:**

1. **Regarding Theorem 1 and Equations (4) and (5):**
   If my understanding is correct, you take for granted that Equations (4) and (5) have solutions and that their marginals are given by $\rho_0$ and $\rho_1$. If this is the case, could you clarify what is novel in **Theorem 1**?
   Given that we already know the marginal of the solution corresponds to $\rho_0$, and assuming $\rho_0$ is defined as in line 145 (i.e., concentrated on a finite set of points $\{X_i\}$), it logically follows that $Z_0$ must coincide with one of the $X_i$.
   What, then, does **Theorem 1** contribute beyond this observation?

2. **Regarding the claim on lines 175–176:**
   Is my understanding correct that the claim on lines 175–176 implies the formula for $b^\star$ can *only* be derived when $\rho_0$ is concentrated on a finite set?
   If so, could you explain why this restriction is necessary?
   To my knowledge, at least in the case where $\gamma = 0$, similar formulas for $b^*$ are available for a much broader class of distributions, not limited to discrete ones.

---

### Official Review · Reviewer_sjbn · 2025-11-01

**Soundness:** 4
**Presentation:** 2
**Contribution:** 3
**Rating:** 4
**Confidence:** 4

**Summary:**

This paper provides a theoretical investigation into the behavior of generative models under a finite training set, focusing specifically on the stochastic interpolation framework. The authors derive closed-form expressions for the optimal velocity field and score function given a finite number of training samples. The analysis reveals that in an ideal (error-free) setting, the deterministic generation process exactly recovers the training samples, while the stochastic generation process yields training samples corrupted by additive Gaussian noise. Furthermore, the paper extends this analysis to the practical setting involving model estimation errors, introducing formal definitions for "overfitting" and "underfitting" in this context. The theoretical results suggest that generated samples in this scenario can be viewed as a mixture of training samples, uniform noise (from estimation error), and Gaussian noise (from stochasticity). The paper's theoretical claims are supported empirically by 2D data visualizations and downstream tasks (classification, contrastive learning) on datasets like MNIST, CIFAR-10, and ImageNet, providing a theoretical foundation for understanding memorization and generalization in diffusion and flow-based models under limited data.

**Strengths:**

- The core strength of this work lies in its rigorous theoretical contributions. It successfully derives closed-form expressions for the optimal velocity field and score function within the SI framework under the practical and important finite-sample assumption. The resulting softmax structure of the weights provides a clear and intuitive understanding of the "attractor effect" pulling the generation trajectory towards the training data points. This is followed by clean, limiting conclusions for deterministic (exact sample recovery) and stochastic (sample + Gaussian noise) generation in the ideal case.
- The paper does not stop at the ideal error-free setting. It systematically characterizes the impact of estimation errors on the generation process. This analysis provides actionable theoretical guidance for practitioners, linking generation outcomes to the properties of the error and the relative decay rates of the interpolation schedules (e.g., $\gamma(t)$ and $\beta(t)$), which is valuable for practical understanding and parameter tuning.

**Weaknesses:**

- The analysis is confined to the stochastic interpolation framework (and the diffusion models it encompasses). It remains an open question whether these conclusions generalize to other prominent generative model families, such as VAEs, GANs, and Autoregressive models. Furthermore, several core conclusions rely on idealized assumptions:
(1) The standard setup assumes $\rho_1$ is a Gaussian distribution.
(2) Specific technical conditions (e.g., $\zeta(t) \lesssim B(t)$) are required for some results.
(3) The analysis in Sec 3.2 depends on strong, formal assumptions about the error term $\epsilon(z,t)$. For instance, Corollary 1 assumes the error is "uniformly bounded" , while Corollary 3 assumes it is "inversely proportional to the density." These assumptions, while facilitating theoretical derivation, may not hold in practice, and their robustness for industrial-scale models or multi-modal generation is yet to be verified.
- The empirical support for the core theoretical claims (especially regarding the impact of different error bounds) relies heavily on 2D "toy" data visualizations. While the downstream tasks on small- to medium-scale datasets (MNIST, CIFAR-10) are illustrative, it remains unclear whether the insights derived from these low-dimensional error structures generalize to the complex, high-dimensional data manifolds encountered in state-of-the-art image generation. The paper would be strengthened by validation on larger-scale or conditional generation settings.

**Questions:**

It appears that this submission may not fully comply with the official ICLR 2026 LaTeX style and template requirements. In particular, the section headings, theorem statements, and corollaries are not displayed in boldface as specified in the ICLR formatting guidelines. This results in weak visual separation between structural elements, thereby reducing readability and making it difficult to distinguish different logical components of the paper. Could the authors confirm whether the manuscript was prepared using the correct ICLR 2026 LaTeX style files and templates?

---

### Note · Authors · 2025-11-12

I have read and agree with the venue's withdrawal policy on behalf of myself and my co-authors.